# Amaranth Leaves and Skimmed Milk Powders Improve the Nutritional, Functional, Physico-Chemical and Sensory Properties of Orange Fleshed Sweet Potato Flour

**DOI:** 10.3390/foods8010013

**Published:** 2019-01-04

**Authors:** Gaston Ampek Tumuhimbise, Gerald Tumwine, William Kyamuhangire

**Affiliations:** School of Food Technology, Nutrition and Bioengineering, College of Agricultural and Environmental Sciences, Makerere University, P.O Box 7062 Kampala, Uganda; tgerald111@gmail.com (G.T.); wkyamuhangire@gmail.com (W.K.)

**Keywords:** functional properties, orange fleshed sweet potato, vitamin A, porridge, skimmed milk

## Abstract

Vitamin A deficiency (VAD) and under nutrition are major public health concerns in developing countries. Diets with high vitamin A and animal protein can help reduce the problem of VAD and under nutrition respectively. In this study, composite flours were developed from orange fleshed sweet potato (OFSP), amaranth leaves and skimmed milk powders; 78:2:20, 72.5:2.5:25, 65:5:30 and 55:10:35. The physico-chemical characteristics of the composite flours were determined using standard methods while sensory acceptability of porridges was rated on a nine-point hedonic scale using a trained panel. Results indicated a significant (*p* < 0.05) increase in protein (12.1 to 19.9%), iron (4.8 to 97.4 mg/100 g) and calcium (45.5 to 670.2 mg/100 g) contents of the OFSP-based composite flours. The vitamin A content of composite flours contributed from 32% to 442% of the recommended dietary allowance of children aged 6–59 months. The composite flours showed a significant (*p* < 0.05) decrease in solubility, swelling power and scores of porridge attributes with increase in substitution levels of skimmed milk and amaranth leaf powder. The study findings indicate that the OFSP-based composite flours have the potential to make a significant contribution to the improvement in the nutrition status of children aged 6–59 months in developing countries.

## 1. Introduction

Under nutrition affects millions of people globally, especially in developing countries [1]. The cause of under nutrition is mainly an inadequate nutrient intake or absorption to cover needs for energy, growth and to maintain a healthy immune body system. Micronutrient deficiencies are a form of undernutrition and occur when the body lacks one or more micronutrients such as iron, iodine, zinc, vitamin A or folate. These deficiencies usually affect growth and immunity but some cause specific clinical conditions such as anaemia (iron deficiency), hypothyroidism (iodine deficiency) or xerophthalmia (vitamin A deficiency). Vitamin A [2] and iron deficiencies [3] are the major public health concerns in resource poor communities. Macronutrient deficiencies are also common and usually occur as protein energy malnutrition. The persistent high levels of macro and micronutrient deficiencies in developing countries are attributed to the dependence on plant based foods and lack of nutrient diversity [4]. Plant based foods are relatively cheaper and can be afforded by most households in developing countries. However, they have a relatively lower protein quality and limited nutrient bioavailability. Inadequate intake of quality protein and micronutrients such as iron and vitamin A [2] might have contributed to the widespread of macro and micronutrient malnutrition manifested in children aged 6–59 months [5]. Many programs have been implemented to reduce vitamin A, iodine [6] and iron deficiencies in developing countries. One of the programs that has been implemented to reduce micronutrient deficiencies is the use of bio-fortified sweet potatoes such as orange fleshed sweet potatoes (OFSP) and beans respectively [7,8]. Indeed in many areas, OFSP have been used in the formulation of complementary diets due to their high content of naturally bio-available β-carotene [9]. Studies have indicated that OFSP flour is rich in β-carotene (100–1600 mg RAE/100 g for varieties in Africa) [10], energy (293 to 460 kJ/100 g) [11] and significant amounts of iron, zinc and manganese [12]. Although OFSP and its products may have many positive attributes and is cheaper than other crops, it is limited in other micronutrients such as calcium, sodium, potassium, phosphorus and quality proteins [13]. Therefore, OFSP alone cannot be adequate in combating the different types of nutrient deficiencies afflicting the vulnerable communities in developing countries. Thus, there is a need to enhance the macro and micronutrient profile of OFSP using locally available foods. Green leafy vegetables such as amaranth have been documented to contain essential micronutrients such as β-carotene, vitamin C, iron, calcium, zinc and proteins [14].

Amaranth leaves are considered as one of the principal leafy vegetables in tropical areas with high annual production [15]. However, they are mainly used as salads and sauces by adults in most areas [16]. On the other hand, skimmed milk powder is an excellent source of proteins (34 to 37%) and is rich in calcium (1257 mg/100 g) [17]. Milk protein is the source of all the essential amino acids with high protein digestibility [18]. Therefore, addition of skimmed milk and amaranth leaves powder to orange fleshed sweet potato flour could be a better option to provide a better overall essential amino acid balance and micronutrients. This could help to overcome the global protein calorie and micronutrient malnutrition challenges respectively. The aim of this study was therefore to develop a nutrient enhanced OFSP-based composite flour incorporating skimmed milk and amaranth leaf powders that is suitable for children aged 6–59 months.

## 2. Materials and Methods

### 2.1. Source of Raw Materials and Laboratory Reagents

Orange fleshed sweet potatoes (NASPOT 13 variety, maturity; 5 months) and amaranth leaves were obtained from the National Crop Resources Research Institute (NaCRRI), Namulonge, Uganda. Skimmed milk powder was purchased from Pearl Dairies, Mbarara District, Uganda. All the materials were delivered to the laboratory at the School of Food Technology, Nutrition and Bioengineering, Makerere University for further processing. Laboratory reagents were purchased from Neo Faraday Laboratory Supply, Kampala, Uganda.

### 2.2. Preparation of OFSP Flour and Amaranth Leaves Powder

The OFSP roots were manually washed, peeled, blanched in hot water in a water bath (Grants Instrument Ltd., Shepreth, UK) maintained at 90 °C for 2 min [19] and cut into thin pieces using a hand grater with holes of diameter of 0.6 cm. Amaranth leaves were washed with potable water to remove surface soil. The amaranth leaves were dipped in 5% saline solution for 15 min. The amaranth leaves and grated OFSP were separately spread on solar drier trays and dried in a locally made solar drier (KENTMARK Ltd, Kampala, Uganda; tunnel dryer maximum, visqueen UV 4; 6250062, temperature 56 ± 2 °C) for 24 h. The dry OFSP pieces and amaranth leaves were separately milled into fine powders using a locally fabricated hammer mill. The OFSP flour and amaranth leaves powder were separately packaged in aluminum laminated packages [20] and stored in the freezer before they were mixed to produce composite flours. 

### 2.3. Composite Flour Preparation

The five different combinations of orange fleshed sweet potato, amaranth leaves and skimmed milk powders (Table 1) were determined with Nutri-survey software and used in the composite flour. The selection of proportions of each ingredient used in composite flour was based on the nutritional requirements of children aged 6–59 months and took into consideration the effect of amaranth leaves powder on the color of the resultant porridge [21,22]. Orange fleshed sweet potato flour was blended with amaranth leaves and skimmed milk powder by using a mixer (Lilaram Manomal and Sons, Vadodara, Gujarat, India). The composite flour samples were packaged in aluminum laminated packages and stored in plastic buckets at room temperature (25 ± 5 °C). The OFSP-based composite flour samples were randomly given codes GT1, GT2, GT3 and GT4 (Table 1).

### 2.4. Nutrient Composition of OFSP-Based Composite Flours

The proximate composition of OFSP and OFSP-based composite flours were carried out according to AOAC official method for nutrient analysis [23]. Moisture content was determined by the oven method; protein content was determined by Kjeldahl method (nitrogen content multiplied by 6.25); fat content was determined by using petroleum ether extraction; and crude fiber was determined by digesting defatted samples with diluted acid (1.25%) sulfuric acid solution for 30 min at boiling point followed by digestion with 1.25% sodium hydroxide solution for the same duration. The carbohydrate component was determined by difference while the energy content was determined by using the Atwater factor (carbohydrate and protein values were each multiplied by 4 kcal/g, whereas fat values were each multiplied by 9 kcal/g).

The amount of calcium, zinc, iron and phosphorus in the composite flours was measured using an atomic absorption spectrophotometer (AAS) [24]. About 5 g of flour was placed in a previously weighed porcelain crucible and heated. The resulting white ash was weighed, dissolved in 10 mL of 1:1 nitric acid (prepared by dissolving 5 mL of nitric acid in 5 mL of distilled water), filtered into a 50 mL volumetric flask and diluted with distilled water to the 50 mL mark. The solutions were then taken to an atomic absorption spectrophotometer (Atomic Absorption Spectrophotometer, Shelton, CT, USA) and the absorbance read at 470 nm which was later used to determine the concentrations of calcium, zinc, iron, magnesium and copper. Standard stock solutions of calcium, zinc, iron, magnesium and copper were also prepared from AAS grade chemicals by appropriate dilution. Calibration curves were obtained by plotting the concentration against the absorbance for the calcium, zinc, iron, sodium, and potassium measurements. Calibration equations were derived and concentrations of calcium, zinc, iron and phosphorus were expressed as mg/100 g.

The vitamin A (RAE) content of flours was estimated by acetone-petroleum ether extraction followed by spectrophotometric measurement and total carotenoids divided by 12 according to the modified Rodrigues-Amaya and Kimura method of total carotenoids analysis [25]. Extraction of carotenoids was performed by grinding of composite flours in a mortar using a pestle, filtration through a filter funnel filled with glass wool and separation from acetone to petroleum ether. The petroleum eluent adjusted to a specific volume was read at 450 nm in a spectrophotometer (ThermoFisher Scientific, Waltham, MA, USA) for the concentration of total carotenoids. Vitamin A (RAE) was obtained by dividing total carotenoids by 12 and results expressed as micrograms per 100 g of dry weight (µg/100 g). All analyses were carried out in triplicate and measured on a dry weight basis.

### 2.5. Determination of Physico-Chemical and Functional Properties Of OFSP-Based Composite Flours

The bulk density, water and oil absorption capacities of the OFSP-based composite flours were determined as described by reference [26]. About one gram of flour was mixed with 10 mL of distilled water and 10 mL of oil respectively in a test tube and vortexed for about 5 min. The samples were allowed to stand at 30 °C for 30 min and then centrifuged at 10,000 rpm for 30 min using centrifuge (ThermoFisher scientific, Waltham, MA, USA). The volume of supernatant in a graduated cylinder was then noted. Density of water was taken to be 1 g/mL and that of oil to be 0.93 g/mL. The water/oil absorbed (V) was calculated as the difference between the initial water/oil used (V_0_) and the volume of the supernatant obtained after centrifuging (V_1_). Thus V= V_0_ − V_1_ and mass = density × volume. The percentage of water/oil absorbed by the flour was expressed on a % basis. For bulk density, approximately 2 g of the flour (m) was gently introduced into a dry 10 mL graduated cylinder without compacting. The cylinder was carefully tapped to compact the sample. The apparent volume (V) was recorded to the nearest graduated unit. The bulk density of flours was expressed as g/mL.

The swelling power and solubility of the flours were determined according to the method described by reference [27]. About one gram (1 g) of sample was weighed, transferred into a clean dry test tube and added to 50 mL of distilled water. The mixture was vortexed for about 5 min. The resulting slurry was heated at 60 °C for 30 min in a water bath (Grants Instrument Ltd., Shepreth, UK) while shaking the tubes every after 5 min. The mixture was cooled to room temperature and centrifuged at 2200 rpm for 15 min. Supernatant was carefully removed and starch sediment weighed. About 5 mL of aliquot of the supernatant was taken into pre-weighed dish and dried to a constant weight at 120 °C for 4 h. The residue obtained after drying was taken to be the amount of starch solubilized in water. The swelling power and solubility of the sample were expressed as percentages.

Pasting characteristics of the porridge from the composite flours were determined using a Rapid Visco Analyzer (Perten Instruments AB, Kungens Kurva, Sweden) according to the methods described by reference [28]. Peak viscosity, trough, breakdown, final viscosity, setback, peak time and pasting temperatures were read from the pasting profile with the aid of thermocline for Windows software connected to a computer [29]. The viscosity was expressed in centipoises (cP).

### 2.6. Contribution of Porridge from Composite Flours to Recommended Dietary Allowance (RDA) Of Children Aged 6–59 Months

Percentage contribution to recommended dietary allowance was expressed as a % of RDA.
(1)$$\%\mathrm{RDA}=\frac{X}{Y}\times 100$$
where X is the amount of nutrient analyzed and Y is the RDA for a given nutrient/variable.

### 2.7. Sensory Evaluation of Composite Flour Porridges

Porridges were prepared by separately adding 200 g of OFSP and OFSP-based composite flours in 250 mL of cold water. The resulting paste was added to 550 mL of boiling water and cooked for 15 min with constant stirring. The prepared porridge was kept in coded thermos vacuum flasks. The sensory attributes of porridges were evaluated by thirty (30) trained panelists comprising of students and staff in the School of Food Technology, Nutrition and Bio-Engineering, Makerere University. The ages of panelists ranged from 18 to 45 years and there were 16 females and 14 males. Each panelist sat in an individual booth and was provided with hot porridge samples in plastic disposable cups marked with 3-digit random codes. Each panelist was provided with drinking water to rinse the palate after each taste. The sensory attributes of porridges that were evaluated included general appearance, color, taste, aroma, thickness, and overall acceptability. The attributes were rated on a nine-point hedonic scale (like extremely = 9 to dislike extremely = 1).

### 2.8. Statistical Analysis

All experimental determinations were carried out in triplicate and subjected to statistical analysis of variance (ANOVA) using XLSTAT software version 2017 (Addinsoft, New York, NY, USA) to determine variation between means of OFSP-based composite flours for their nutrient composition, sensory, physico-chemical and functional properties. A multiple factor analysis (MFA) was run to determine correlation between sensory attributes of porridges from different OFSP-based composite flours. Significance variation was accepted at *p* < 0.05. The Fisher Least Significant Difference (LSD) test was done to determine the significant difference between the two means of the properties of flours. Experimental results were expressed as the means ± standard deviations (SD).

## 3. Results and Discussion

### 3.1. Nutrient Composition of Orange Fleshed Sweet Potato-Based Composite Flours

The moisture, ash, protein, fat, carbohydrate, fiber and energy contents of OFSP-based composite flours on dry weight basis are presented in Table 2. The moisture content of OFSP and OFSP-based composite flours ranged from 5.4 to 5.9%. The moisture content of OFSP and OFSP-based composite flours was slightly higher than the moisture content (<5%) recommended by Codex standards for complementary foods but below the critical moisture (12%) content for flours. The low moisture content of the OFSP and OFSP-based composite flours is attributed to proper drying and handling. Therefore, the OFSP-based composite flours would be stable on the shelves for longer periods due to their low moisture contents. On the other hand, a moisture content ranging from 6.9 to 10.9% in different varieties of OFSP flours was reported by reference [30]. This implies that the OFSP-based composite flours in this study would be more shelf-stable than those reported by reference [30].

The carbohydrate content of OFSP-based composite flours significantly (*p* < 0.05) decreased from 86.0 to 67.8%. The decrease in carbohydrate content is attributed to the dilution effect of skimmed milk (49.5–52.0%) [31] and amaranth leaf powders (28.2%) [32], which are low in total carbohydrates. However, the carbohydrate content of the OFSP-based composite flours is within the range (45 to 65%) recommended for infant feeding, making it suitable for use in the preparation of porridges for children aged 6–59 months. According to Amagloh and Coad [33], carbohydrate content of sweet potato, skimmed milk powder and maize based complementary foods was in the range 50.25 to 58.92%, which are lower than those reported in this study. A similar trend (64.8 to 57.1%) was also reported by Nkesiga and Okafor [34] with the addition of amaranth leaves powder in yellow maize flour at 20% substitution level. The higher carbohydrate content reported in this study could be due to differences in the proportions of ingredients used.

There was no significant difference between the energy content of OFSP and OFSP-based composite flours (*p* > 0.05) (Table 2). This might be explained by the fact that the ingredients added to OFSP did not have a lot of carbohydrates and therefore could not significantly alter the energy content of OFSP flour [35]. Findings in this study indicated a higher energy content compared to 350–360 kcal/100 g reported by [34] in yellow maize flour supplemented with 20% amaranth leaf powder. The higher energy content reported in this study may be explained by the high fat and carbohydrate contents recorded (Table 2). The energy content of OFSP-based composite flours is approximately half the total energy required for healthy breastfed infants; 615 kcal/day from 6 to 8 months of life, 686 kcal/day from 9 to 11 months and 894 kcal/day from 12 to 23 months [36]. Therefore, the OFSP-based composite flours are suitable for use in making porridges for children aged 6–59 months.

Study findings further indicated that addition of skimmed milk and amaranth leaf powder significantly (*p* < 0.05) increased the ash content of OFSP-based composite flours from 2.7 to 5.3%. The significant increase in the ash content in OFSP-based composite flours may be attributed to addition of amaranth leaf powders because they are reported to be rich in minerals (10.6% ash) [32]. Findings in this study are in agreement with those of Nkesiga and Okafor [34] who reported an increase in ash content in yellow maize flour from 1.3 to 4.6% with addition of 20% amaranth leaf powder. Thus, the OFSP-based composite flours would contribute to the recommended dietary allowances (RDA) of minerals required by children aged 6–59 months.

The protein content of OFSP-based composite flours significantly (*p* < 0.05) increased from 4.1 to 19.9%. The protein content of GT3 and GT4 was higher than the protein value (15%) stipulated in the Codex standard of complementary foods. Therefore, GT3 and GT4 comply with the permitted levels (15%) of formulated complementary foods [37]). This observation could be due to the fact that blending of two or more plant and animal-based food materials increases the nutrient density of the food product [38]. Therefore, the addition of skimmed milk (34–37% protein) and amaranth leaves (32.5% protein) increased the protein content of OFSP-based composite flours. According to Mahmoud and Anany [39], an increase in the protein content (17.9%) of a complementary food formulated from rice, fib beans, sweet potato flour, and peanut oil was reported. This was probably due to incorporation of legumes (fib beans and peanut), which are rich in proteins.

There was a significant increase (*p* < 0.05) between the fat content of OFSP-based composite flours. The increase in fat content of OFSP-based composite flour is attributed to increase in levels of skimmed milk powder added to OFSP flour. Findings from this study showed a lower fat content (0.7 to 1.4%) than that (3.87 to 5.17%) reported by Tadese et al. [40] in flat-bread prepared from blends of maize and OFSP flours. This is because maize flour has higher fat content (6.95%) [40] than skimmed milk powder (1.5%). The low-fat content of OFSP-based composite flours may be better for longer storage of the OFSP-based composite flours if properly packaged and stored in areas with low humidity and temperature.

The crude fiber content of OFSP-based composite flours significantly (*p* < 0.05) increased from 1.2 to 3.2%. This is attributed to higher levels of amaranth leaf powder added that is reported to have higher fiber content (18.11%) [32] than OFSP flour (2.57%) [24]. Findings from this study are in agreement with those reported by Beswa et al. [32], who observed an increase in the crude fiber content of extruded pro-vitamin A bio-fortified maize snacks with addition of 1 and 3% amaranth leaf powder. The crude fiber content of the OFSP-based composite flours was within the recommended Codex Standards (5%) for complementary foods. Therefore, the composite flours are suitable for use in complementary feeding of children aged 6–59 months.

The results for mineral and vitamin A (µg RAE) content of OFSP-based composite flours are presented in Table 3. The vitamin A (µg RAE) content of OFSP and OFSP-based composite flours decreased from 1989.8 to 145.7 µg RAE/100 g with increase in the substitution levels of amaranth leaves and skimmed milk powders. The decrease in vitamin A (µg RAE) content of OFSP flour was significantly (*p* < 0.05) different from that of OFSP-based composite flours. This could be attributed to the dilution effect due to addition of skimmed milk powder that has low levels of vitamin A (µg RAE). Findings from this study showed that vitamin A concentrations were higher than those reported by Amagloh and Coad [33] in orange fleshed sweet potato-based infant food (226.24 µg RAE/100 g). In addition, Kidane et al. [41] reported 1924 µg RE/100 g in orange fleshed sweet potato flour, which was slightly lower than that reported in this study. The differences in vitamin A (µg RAE) observed in this study and those reported in other studies [33,41] are attributed to drying temperatures and time.

Study findings also indicated a significant (*p* < 0.05) increase in iron (4.8 to 97.4 mg/100 g), calcium (45.5 to 670.2 mg/100 g) and phosphorus (69.2 to 388.3 mg/100 g). The increase in iron, calcium and phosphorus content of OFSP-based composite flours is attributed to the addition of amaranth leaves powder because they are reported to be rich in these minerals [32]. In addition, the high calcium content reported in this study is attributed to the addition of skimmed milk powder (1257 mg/100 g) [31].

### 3.3. Contribution of Energy and Protein Content Of Porridge Prepared from 200 G Of OFSP-Based Composite Flours in 800 mL Of Water Towards RDA for Children Aged 6–59 Months

Table 4 shows the contribution of OFSP-based composite flours to the RDAs of energy and protein for children aged 6–59 months. The study findings indicated that the protein contribution to the RDA reduced with an increase in the age of children. For children 0.5–1 year, the porridge from OFSP-based composite flours contributes 86.4 to 142.1% of the RDA for protein but it contributes only 50.4 to 82.9% for children aged 4–6 years. The energy contribution was 45.5 to 44.6% for children 0.5–1 year and 21.5 to 21.1% for children 4–6 years (Table 4).

The high contribution of the OFSP-based composite porridges to RDAs for protein and energy are due to their reported high concentrations in the composite flours (Table 2). In addition, the contributions of protein that are above the RDA are non-toxic to the body because it was slightly above the protein requirement [32]. However, it is recommended that protein intake should not be more than twice the RDA for protein [37]. Reduction in the contribution of energy and protein to the RDA with an increase in age is due to an increase in the body’s needs during growth. For example, energy is needed for metabolic activities and body maintenance while protein is needed for growth and development in children. In order to meet the protein and energy RDAs of the older children, an intake of more than 100 mL of the OFSP-based composite porridge is recommended.

### 3.4. Contribution (%) of Calcium, Iron and Vitamin a Of Porridge from OFSP-Based Composite Flours Towards the RDA for Children Aged 6–59 Months

Table 5 shows the calcium, iron and vitamin A contribution (%) of porridge from OFSP-based composite flours to the recommended dietary allowances for children aged 6–59 months. Findings from this study showed that the porridge from composite flours contributed more than 100% of the required iron. However, the porridge from OFSP flour only contributed 48% of the RDA for iron in children aged 6–59 months. The results further indicate that the mean calcium contributions of composite flours were between 45.9% and 95.7% of the RDA for children aged 6–59 months. Findings further show that the vitamin A contribution was above 100% in OFSP, GT1 and GT2 while those of GT3 and GT4 were below 100%. The high contributions of iron zinc, calcium and vitamin A were due to high concentrations as indicated in Table 3. Iron and vitamin A are non-toxic in the body [37,42] and therefore their high contribution levels in the OFSP-based composite flour have no health concern. Therefore, adoption of the OFSP-based flours and their proper preparations may greatly contribute to the reduction of mineral and vitamin A deficiencies among children aged 6–59 months.

### 3.5. Physico-Chemical and Functional Properties of Orange Fleshed Sweet Potato-Based Composite Flours

Table 6 presents the physico-chemical and functional properties of orange fleshed sweet potato-based composite flours on dry weight basis. The functional properties determine the application and use of food materials for various food products. The decrease in solubility of flours was not significant (*p* = 0.423) while swelling power significantly (*p* = 0.048) decreased from 0.9 to 0.5%. The non-significant decrease in solubility of OFSP-based composite flours is attributed to the dilution effect of sugars in the flours by addition of skimmed milk and amaranth leaves powder that have lower sugar content than OFSP flour. According to reference [43], high sugar content favors the formation of hydrogen bonds, increasing solubility. Therefore, the low solubility of OFSP-based composite flours is due to low sugars. As such, the developed composite flours are less soluble in water due to low sugars but would still be soluble due to high protein content that exposes hydrophilic groups during porridge making. The study findings are in agreement with those of reference [44], who reported a solubility of 3.12% in orange flesh sweet potato-sorghum-soy blend at a ratio of 40:40:20.

The swelling power indicates the degree of water absorption of the starch granules in the flour during heating [45]. As a result of water absorption and heat, starch granules swell resulting in a viscous paste. There were no significant (*p* > 0.05) differences in the swelling power among the composite flours but significant (*p* = 0.048) difference was observed between composite flours and OFSP flour (control). The increasing levels of amaranth leaves and skimmed milk powders decreased the swelling power of composite flours. This is probably due to reduction in the number of starch granules due to lower carbohydrate content (Table 2) as a result of the addition of skimmed milk and amaranth leaf powders. The low swelling power of the composite flours makes them suitable for use in the preparation of gruels used as weaning foods, as they will result in porridges of low viscosity desirable for children due to the low volumes of their stomachs. Findings from this study are consistent with those of reference [46] that reported a decrease in swelling power in sweet potato-based composite flour with an increasing amount of soybean flour being added.

Water absorption capacity is the ability of flour to absorb water and swell, for improved consistency in food. It is desirable for food systems to improve yield and consistency and to give body to the food. The water absorption capacity (WAC) of OFSP and OFSP-based composite flours significantly (*p* < 0.05) decreased from 62.8 to 58.0%. On the other hand, there was no significant (*p* > 0.05) decrease in WAC between samples GT2, GT3 and GT4. The high WAC recorded for the OFSP flour could be due to its small and uniform particle size, giving a higher surface area and high capillarity in the flour [47]. The values of the water absorption capacity obtained for the flours correspond with the swelling power and solubility. This implies that the low WAC of the OFSP-based composite flours obtained in this study will be desirable for making thinner gruel with a high caloric density per unit value. The oil absorption capacity (OAC) of OFSP and OFSP-based composite flours significantly (*p* < 0.05) increased from 25.4 to 73.5% (Table 6). Oil absorption is important because oil acts as a flavor retainer and increases the mouth feel of foods, improves palatability and extends the shelf-life of foods, especially in bakery or meat products where fat absorptions are desired [48]. The increase in OAC in OFSP-based composite flours is attributed to the high protein content due to the addition of skimmed milk and amaranth leaves powders. The high protein content of composite flours enhanced hydrophobicity by exposing more polar amino acids to the fat. This observation is consistent with the reports of reference [46], who observed an increase in OAC of composite flours prepared by blending sweet potato flour with maize flour, soy bean flour and xanthan gum from 2.03 to 2.2 g/g. This is probably due to the addition of skimmed milk and amaranth leaves powders that are rich in proteins. The high OAC of the composite flours indicates that the flours could also be used in making bakery products for infants.

The bulk density of the flours ranged from 0.5 to 0.6 g/mL. The bulk densities obtained in this study were insignificantly (*p* > 0.05) very low and this indicates that the flours would be advantageous in the preparation of complementary foods. The study findings are in agreement with those of reference [44] who reported a bulk density of 0.6 g/mL in orange flesh sweet potato, sorghum and soybean blend. Bulk density is a measure of heaviness of a flour sample and this gives an indication that the relative volume of the composite flours in a package will not reduce excessively during storage.

Figure 1 shows the pasting properties of OFSP-based composite flours. The results indicated that OFSP flour recorded the highest peak (1046.5 cP) and final (191.5 cP) viscosities. The peak and final viscosities of the composite flours decreased with increasing levels of substitution of skimmed milk and amaranth leaves powders. The decrease in peak viscosity was from 464.0 to 180.0 cP, while that of final viscosity was from 122.5 to 116.5 cP. The decrease in peak and final viscosities of composite flours compared to OFSP flour is attributed to the high fiber contents of the composites due to addition of amaranth leaves powders. Fiber competes with starch for the limited amount of water available in a food system [49] thus reducing the viscosity. The final viscosity is the change in viscosity after holding cooked starch at 50 °C and it indicates the ability of starch to form a viscous paste or gel after cooking and cooling [43]. The results in Figure 1 indicate that composite flours had lower final and peak viscosities than OFSP flour. This is nutritionally beneficial in infant formulas since a less viscous porridge is a better weaning food for children.

The setback or viscosity of cooked paste is the viscosity after cooling the paste to 50 °C. The extent of increase in viscosity on cooling to 50 °C reflects the retrogradation tendency, a phenomenon that causes the paste to become firmer and increasingly resistant to enzyme attack [50]. It thus has an effect on digestibility. Higher setback values are synonymous with reduced paste digestibility [51], while lower setback during cooling of the paste indicates a lower tendency for retrogradation and subsequently higher digestibility. The low setback values for the OFSP-based composite flours indicate that their pastes would have higher stability against retrogradation than OFSP flour. The lower set back viscosities also imply that the porridge when consumed by children would be easy to digest.

The pasting temperature of OFSP-based composite flours increased from 77.9 to 79.9 °C while the pasting time ranged between 3.7 and 3.8 min with an increase in substitution levels of skimmed milk and amaranth leaf powders. The pasting temperatures were significantly (*p* < 0.05) higher than that of OFSP flour (74.3 °C). This provides an indication of minimum temperature required for cooking the porridge from the flours. The high pasting temperature of OFSP-based composite flours implies that more energy will be required for cooking porridge from OFSP-based composite flours than for flour from OFSP.

### 3.6. Sensory Acceptability of Porridges from OFSP-Based Composite Flours

Table 7 presents results from the mean sensory scores of porridge from OFSP-based composite flours. The degree of liking for the general appearance of porridges from composite flours decreased from 7.4 to 3.7 with an increase in the substitution levels of skimmed milk and amaranth leaves powders. Porridge from GT2 had the highest score (7.4) while that from GT4 had the lowest score (3.7). There were no significant (*p* > 0.05) differences in the scores for the appearance of porridge from GT1 and 633 OFSP flours, GT2 and OFSP flour. This could be attributed to the low levels of amaranth leaves powder added. However, significant (*p* < 0.05) differences were observed between GT1, Gt2, GT3 and GT4 (Table 7). This is attributed to the increased levels of amaranth leaves powder added to the OFSP flour. The scores for the color of porridges from composite and OFSP flours followed the same trend as that of general appearance. This is probably because color is one of the attributes assessed under appearance.

The scores for the aroma of porridges from composite flours ranged between 3.9 and 6.7. Porridge from GT2 had the highest score while GT4 had the lowest score. Significant differences in the scores were noted between GT2, GT3 andGT4 then OFSP flour and GT4. This is attributed to the increase in the levels amaranth leaves powder added. A similar trend was also observed in the scores of taste for the porridges. The scores for thickness of porridges from composite flours ranged from 5.9 to 6.9. There were significant (*p* < 0.05) differences in the scores of thickness for porridges from GT2 and GT4. The overall acceptability expresses how the consumer or the panelist generally accepts the product. It was observed that porridge from GT2 was the most accepted (6.8) while that from GT4 was the least accepted (4.6). The high score for the overall acceptability of porridge from GT2 could be due to the familiarity of taste, aroma and color. Findings from this study were in agreement with those reported by other researchers [40] whereby overall acceptability scores of 5.72 to 6.96 in porridges from orange flesh sweet potato, sorghum and soybean blend were recorded.

### 3.7. Correlation between Sensory Attributes of Porridges from OFSP-Based Composite Flours

Figure 2 shows the correlation between the sensory attributes of porridges from OFSP-based composite flours. The map shows that the aroma is highly related to general appearance and it is correlated with the first factor (F1). It can also be confirmed that the general appearance, color and aroma are highly correlated with the first axis. It is also observed that all the sensory attributes are spread in the two of the four quadrants. On the other hand, the second factor (F2) is highly correlated with overall acceptability. Sample GT2 has the highest coordinate on the first axis and is highly related to the second factor (Figure 2), which is highly related to taste and overall acceptability. Therefore, sample GT2 was the most highly accepted, likely due to the low levels of amaranth leaves powder. Furthermore, GT1 is in the direction of color and general appearance. Color and general appearance being the most important factors in determining acceptability of the food product would confirm that sample GT1 was most preferred in terms of these two attributes. This is attributed to very low levels of amaranth leaves powder added that might have not imparted a significant change in the color of OFSP flour. In contrast, porridge from samples GT3 and GT4 has the worst ratings. This observation is consistent with what was earlier observed, which is that no sensory attribute was in this quadrant. A zero correlation was observed between thickness and other sensory attributes for all porridges.

## 4. Conclusions

Incorporation of skimmed milk and amaranth leaves powders resulted in nutrient enhanced orange fleshed-based composite flours with improved nutritional, physico-chemical and functional properties. This study showed that production of OFSP flours enriched with amaranth and skimmed milk powders has potential to contribute to the reduction of malnutrition among children aged 6–59 months in developing countries.

## Figures and Tables

**Figure 1 foods-08-00013-f001:**
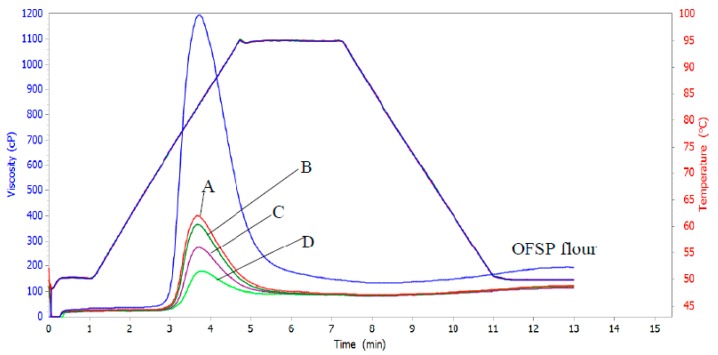
Rapid Visco-Analyzer pasting curves for OFSP and OFSP-based composite flours. Samples A, B, C and D are orange fleshed sweet potato-based composite flours with skimmed milk powder at substitution levels 20%, 25%, 30% and 35% respectively while amaranth leaves powders were 2%, 2.5%, 5% and 10% respectively.

**Figure 2 foods-08-00013-f002:**
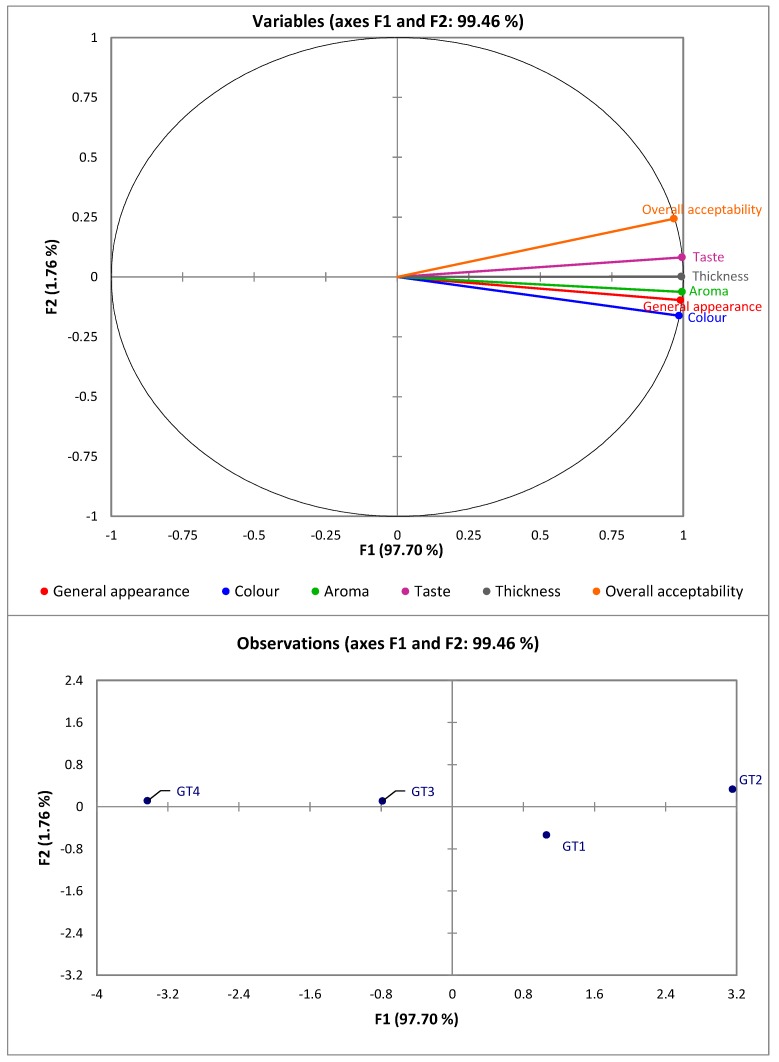
The map/plot showing the correlation between sensory properties of porridges from OFSP-based composite flours.

**Table 1 foods-08-00013-t001:** Different proportions (%) of OFSP, amaranth leaves and skimmed milk powders used in composite flours.

Sample Code	OFSP Flour	Amaranth Leaves Powder	Skimmed Milk Powders
OFSP (Control)	100.0	0.0	0.0
GT1	78.0	2.0	20.0
GT2	72.5	2.5	25.0
GT3	65.0	5.0	30.0
GT4	55.0	10.0	35.0

**Table 2 foods-08-00013-t002:** Proximate (%) and energy (kcal/100 g) composition of orange fleshed sweet potato-based composite flours on dry weight basis (except moisture content).

Sample	Moisture Content	Ash	Crude Protein	Crude Fat	Total Carbohydrates	Crude Fiber	Energy
OFSP	5.8 ± 0.2 ^a^	2.7 ± 0.0 ^d^	4.1 ± 0.3 ^e^	0.4 ± 0.1 ^d^	86.0 ± 0.3 ^a^	1.2 ± 0.4 ^e^	389 ± 0.0 ^a^
GT1	5.7 ± 0.2 ^a^	4.0 ± 0.3 ^c^	12.1 ± 0.5 ^d^	0.7 ± 0.0 ^c^	76.7 ± 0.8 ^b^	1.5 ± 0.0 ^d^	387 ± 0.1 ^a^
GT2	5.7 ± 0.3 ^a^	4.3 ± 0.1 ^c^	13.9 ± 0.4 ^c^	1.1 ± 0.4 ^b^	73.6 ± 0.5 ^c^	2.2 ± 0.3 ^c^	386 ± 0.0 ^a^
GT3	5.4 ± 0.6 ^a^	4.6 ± 0.2 ^b^	17.0 ± 0.6 ^b^	1.1 ± 0.1 ^b^	71.6 ± 1.6 ^c^	2.5 ± 0.0 ^b^	383 ± 0.0 ^a^
GT4	5.9 ± 0.7 ^a^	5.3 ± 0.2 ^a^	19.9 ± 0.4 ^a^	1.4 ± 0.4 ^a^	67.8 ± 0.0 ^d^	3.2 ± 0.4 ^a^	379 ± 0.0 ^a^
*p*-value	0.694	<0.001	<0.001	<0.005	<0.001	<0.05	0.283

Means and standard deviations of triplicate determinations. Means in the same column with different superscripts (^a,b,c,d,e^) are significantly (*p* < 0.05) different. Samples GT1, GT2, GT3 and GT4 are orange fleshed sweet potato-based composite flours with skimmed milk powder at substitution levels 20%, 25%, 30% and 35% respectively while amaranth leaves powders were 2%, 2.5%, 5% and 10% respectively.

**Table 3 foods-08-00013-t003:** Mineral and vitamin A (µg RAE) content of orange fleshed sweet potato-based composite flours on dry weight basis.

Sample	Vitamin A (µg RAE/100 g)	Fe (mg/100 g)	Ca (mg/100 g)	P (mg/100 g)
OFSP flour	1989.8 ± 1.2 ^a^	4.8 ± 0.4 ^e^	45.5 ± 0.4 ^e^	69.2 ± 0.2 ^e^
GT1	1447.3 ± 1.1 ^b^	19.6 ± 0.6 ^d^	321.2 ± 0.2 ^d^	253.7 ± 0.2 ^d^
GT2	563.8 ± 0.4 ^c^	24.5 ± 0.1 ^c^	394.3 ± 0.4 ^c^	299.4 ± 0.9 ^c^
GT3	343.9 ± 0.2 ^d^	48.8 ± 0.1 ^b^	506.2 ± 0.1 ^b^	345.2 ± 0.4 ^b^
GT4	145.7 ± 1.4 ^e^	97.4 ± 0.2 ^a^	670.2 ± 0.3 ^a^	388.3 ± 0.4 ^a^
*p*-value	<0.001	<0.001	<0.0001	<0.001

Means and standard deviations of triplicate determinations. Means in the same column with different superscripts (^a,b,c,d,e^) are significantly (*p* < 0.05) different. Samples GT1, GT2, GT3 and GT4 are orange fleshed sweet potato-based composite flours with skimmed milk powder at substitution levels 20%, 25%, 30% and 35% respectively while amaranth leaves powders were 2%, 2.5%, 5% and 10% respectively.

**Table 4 foods-08-00013-t004:** Contribution (%) of energy and protein content of porridge from 200 g of OFSP-based composite flours in 800 mL of water towards RDA for children aged 6–59 months.

Variable	Age Group (years)	RDA *^a^*	Contribution (%) of OFSP-Based Composite Flours to RDA				
			OFSP	GT1	GT2	GT3	GT4
Energy (kcal/day)	0–0.5	650	59.9	59.5	59.4	58.9	58.3
	0.5–1	850	45.8	45.5	45.4	45.1	44.6
	1–3	1300	29.9	29.8	29.7	29.5	29.2
	4–6	1800	21.6	21.5	21.4	21.3	21.1
Protein (g/day)	0–0.5	13	31.5	93.1	106.9	130.8	153.1
	0.5–1	14	29.3	86.4	99.3	121.4	142.1
	1–3	16	25.6	75.6	86.9	106.3	124.4
	4–6	24	17.1	50.4	57.9	70.8	82.9

Samples GT1, GT2, GT3 and GT4 are orange fleshed sweet potato-based composite flours with skimmed milk powder at substitution levels 20%, 25%, 30% and 35% respectively while amaranth leaves powders were 2%, 2.5%, 5% and 10% respectively. *^a^* Food and Nutrition Board, (1989).

**Table 5 foods-08-00013-t005:** Contribution (%) of calcium, iron and vitamin A of porridge from OFSP-based composite flours towards the RDA for children aged 6–59 months.

Sample	Contribution to RDA		
	Ca	Fe	Vitamin A
OFSP flour	6.5	48.0	442
GT1	45.9	196.0	322
GT2	56.3	245.0	125
GT3	72.3	488.0	76
GT4	95.7	974.0	32
RDA (mg/100 g)	700.0	10.0	0.45

Samples GT1, GT2, GT3 and GT4 are orange fleshed sweet potato-based composite flours with skimmed milk powder at substitution levels 20%, 25%, 30% and 35% respectively while amaranth leaves powders were 2%, 2.5%, 5% and 10% respectively. The recommended levels of the nutrients considered adequate for most healthy children aged 6–59 months [32].

**Table 6 foods-08-00013-t006:** Physico-chemical and functional properties (%) of orange fleshed sweet potato-based composite flours on dry weight basis.

Sample	Solubility (%)	Swelling Power (%)	Water Absorption Capacity (%)	Oil Absorption Capacity (%)	Bulk Density (g/mL)
OFSP flour	2.9 ± 0.3 ^a^	0.9 ± 0.2 ^a^	62.8 ± 0.4 ^a^	25.4 ± 0.4 ^d^	0.6 ± 0.1 ^a^
GT1	2.7 ± 0.3 ^a^	0.7 ± 0.1 ^ab^	59.1 ± 0.1 ^b^	60.7 ± 0.3 ^c^	0.5 ± 0.0 ^b^
GT2	2.3 ± 0.0 ^a^	0.6 ± 0.1 ^b^	58.5 ± 0.5 ^c^	60.5 ± 0.1 ^c^	0.6 ± 0.0 ^a^
GT3	2.1 ± 1.3 ^a^	0.5 ± 0.0 ^b^	58.0 ± 0.1 ^c^	68.1 ± 0.7 ^b^	0.6 ± 0.1 ^a^
GT4	1.5 ± 0.8 ^a^	0.5 ± 0.1 ^b^	58.0 ± 0.5 ^c^	73.5 ± 0.7 ^a^	0.6 ± 0.1 ^a^
*p*-value	0.423	0.048	< 0.001	< 0.001	0.017

Means and standard deviations of triplicate determinations. Means in the same column with different superscripts (^a,b,c,d^) are significantly (*p* < 0.05) different. S GT1, GT2, GT3 and GT4 are OFSP-based composite flours with skimmed milk powder at substitution levels 20%, 25%, 30% and 35% respectively while amaranth leaves powders were 2%, 2.5%, 5% and 10% respectively.

**Table 7 foods-08-00013-t007:** Sensory acceptability of porridges from OFSP-based composite flours.

Sample Code	General Appearance	Color	Aroma	Taste	Thickness	Overall Acceptability
GT1	6.5 ± 1.6 ^b^	6.2 ± 1.9 ^a^	6.0 ± 1.8 ^ab^	5.8 ± 1.9 ^b^	6.6 ± 1.7 ^a^	5.6 ± 2.3 ^bc^
GT2	7.4 ± 1.3 ^a^	6.8 ± 2.1 ^a^	6.7 ± 1.9 ^a^	6.8 ± 2.0 ^a^	6.9 ± 1.9 ^a^	6.8 ± 1.9 ^a^
GT3	4.9 ± 2.2 ^c^	4.6 ± 2.1 _b_	5.2 ± 1.7 ^b^	5.3 ± 2.0 ^c^	6.4 ± 2.1 ^a^	5.3 ± 2.03 ^bc^
GT4	3.7 ± 2.5 ^d^	3.3 ± 2.4 ^c^	3.9 ± 2.1 ^c^	4.2 ± 2.2 ^d^	5.9 ± 1.8 ^a^	4.6 ± 2.4 ^c^
OFSP Flour	7.1 ± 1.3 ^ab^	6.8 ± 1.9 ^a^	6.0 ± 2.1 ^ab^	5.8 ± 2.2 ^b^	6.8 ± 2.0 ^a^	6.2 ± 1.5 ^ab^
*p*-value	<0.001	<0.001	<0.001	<0.001	0.272	0.001

Means and standard deviations of 30 trained panelists. Means in the same column with different superscripts (^a,b,c,d^) are significantly (*p* < 0.05) different. Samples GT1 GT2, GT3, GT4 and OFSP-based composite flours with skimmed milk powder at substitution levels 20%, 25%, 30%, 35% and 0% respectively while amaranth leaves powders were 2%, 2.5%, 5%, 10% and 0% respectively.

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
