# Peer review of "Amaranth Leaves and Skimmed Milk Powders Improve the Nutritional, Functional, Physico-Chemical and Sensory Properties of Orange Fleshed Sweet Potato Flour"

_foods, 2019, doi:10.3390/foods8010013_

Reviewer 1 Report

Dear Editor,

I am pleased to review the assigned manuscript. Overall, a proposed manuscript looks good to me. Authors really did a good job and presented very nice & relevant literature. The English is generally satisfactory, although there are some places where corrections and/or changes are required. The study does have few major issues. The authors should improve the paper following these suggestions.

·         Abstract need bit of attention and should cover theme of whole manuscript

·         The introduction should be better organized. Some of the sentences are not well structured, should be clarified and rewritten. It is advice to link the story in a better way in an introduction to convey a proper message to readers.

·         Please add few suggested and latest references in background section

-       Concept of double salt fortification; a tool to curtail micronutrient deficiencies and improve human health status.

·         Material and method section is pretty good, please just check the template of the formula and equations used.

·         Result and discussion section really need improvement -  it’s not well written and well organized, few short sentences, without any story – you can focus only on latest reference in discussion. I appreciate, it is really good results, but the way of presentation is poor, please improve in next revised version.

·         The conclusion should be better explained and conclude your whole research.

·         Please recheck the reference style, some of the references are not according to the journal instructions.

I am happy to review the revised version.

Author Response

Dear Reviewer,

Kindly find attached our response to your comments

Thanks you

Dr. Gaston A. Tumuhimbise

Reviewer 2 Report

Abstract

Please change the word physical-chemical to physicochemical.

Level of significance - please use p < .05 instead of ≤ .05.

M&M

Seciton 2.1 - Different font size on line 65, please amend.

Section 2.2 - would be helpful to also have some temperature measurement here during that day if possible? Please add more information on the fine powder size, milling technique, etc.

Section 2.3 - please add information for the solar drier as well.

Section 2.4 - The reference used for the proportion is from nearly 2 decades old publication. Ideally this can be removed or changed to a more recent one. The authors should also justify on why these increments were selected. Table 1 - please also add the abbreviation here. What does GT means?

Section 2.5.1 - please elaborate what methods that was used in this study in relation to reference [22]. 

Section 2.5.2 - font size correction needed

Section 2.6 - Same comment to Section 2.5.1

Section 2.9-2.10 - Numbering isn't adjusted properly, please fix.

Section 2.13 - Add details on the post hoc comparison, this was clearly done for the results.

Results (and Discussion?)

In the results the authors attempted to discuss and had partially discussed some of their results. I'd recommend the authors to make more robust and expand the discussion instead of just showing that their results is in agreement with previous studies (see Section 3.7 - Line 377) 

Table 4. The b headings are missing, please amend.

Table 8. It looks like the sensory evaluation was not done in randomised block design (since the sample has the same codes), if this is the case please add in discussion.

Figure 1. Figure unclear on which sample is which, please fix.

Author Response

Dear Reviewer,

Thank you for reviewing our manuscript. Kindly find our response attached .

Regards,

Gaston A Tumuhimbise

Reviewer 3 Report

The authors presented research results on the nutrient enhanced orange fleshed sweet potato‐based composite flour incorporating skimmed milk and amaranth leaf powders suitable for children 6‐59 months. The authors decided to enhance profile of orange fleshed sweet potato using locally available foods such as amaranth leaves and skimmed milk powders documented to contain essential micronutrients and proteins.

These studies are an extension of existing knowledge. The way in which the results are presented must be changed because some parts of the paper is to general. Methods and way of calculation are traditional. Original research article should not contain so many references that is suitable for the review article. In my opinion the whole part of Materials and Methods and Results and Discussion should be rewritten and more consistently described because in its present form it can not be accepted. Statistical analysis is not described correctly. Therefore, in my opinion the work should be rejected.

Please check manuscript in terms of typos and spelling.

Specific comments are included in the pdf of the article.

Author Response

Dear Reviewer,

Thank you for reviewing our manuscript. Kindly find our response to your comments attached.

Regards,

Gaston A. Tumuhimbise

Round  2

Reviewer 1 Report

Dear Editor, 

I am happy with the revised version but there are still few technical mistakes.

Means and standard deviations 5.8a±0.2 ------------ should be presented in 5.8±0.2a this format, please check the journal guidelines 

Figure 1 is still not fully formatted, readable and visible - please check the submitted pdf.

Best Wishes 

 Author Response

Dear reviewer,

We have recieved your comments and made the necessary changes in the manuscript. The response to the comments is hereby attached

Thanks

Reviewer 2 Report

I'd like to thank the authors for addressing the comment. A significant amount of work has been added to the manuscript.

Section 3.4 - the paragraph alignment needs to be amended

Figure 1. I'm unsure what had happened to the figure, this needs amendment.

Table 7. In sensory evaluation it is a common practice to ensure that all the participants have different codes where the order is also randomised and counterbalanced. So, by having one code for one sample this raises the question whether the authors have done randomisation properly. Additionally, instead of having sample 3-digit code, I'd recommend the authors to change it to the GT codes that the authors have already.

I'd recommend the authors to run a MFA to further summarise their chemical and sensory results to show which sample is the best out of the four formulations.

Author Response

Dear Editor,

Here  is our response to the queries raised

Thank you

Gaston

Reviewer 3 Report

Statistical analysis is not described correctly by the authors. Therefore, in my opinion the work should be revised.

Specific comments are included in the pdf of the article.

Author Response

Dear Reviewer,

We have carefully responded to the comments raised. Our response is hereby attached

Thanks

Round  3

Reviewer 2 Report

The PCA needs adjustments in terms of the coefficients as the vectors are not aligned properly to the product loadings. It is not recommended for the authors to actually merge all the chemical data together with the sensory data in one PCA analysis. Multiple Factor Analysis (MFA) is more appropriate in this case. I'm aware that the authors are using XLSTAT, please refer to this tutorial on how to run the MFA: https://help.xlstat.com/customer/en/portal/articles/2062254-multiple-factor-analysis-mfa-in-excel-tutorial?b_id=9283

Please add the details of the multivariate approach on Section 2.8.

Author Response

Dear Reviewer

Kindly find attached our response to the third set of your comments.

Thank you

Reviewer 3 Report

In my opinion article can be accepted after minor text corrections.

Author Response

Dear Reviewer

Kindly find attached our response to your third set of comments.

Thank you
